# An Approach for Using a Tensor-Based Method for Mobility-User Pattern Determining

Ivan P. Ashaev, Ildar A. Safiullin, Artur K. Gaysin *, Adel F. Nadeev and Alexey A. Korobkov

Radioelectronics and Telecommunication Systems Department, Kazan National Research Technical University Named after A.N. Tupolev-KAI, K. Marx Str. 10, 420111 Kazan, Russia; ashaev.ip@mail.ru (I.P.A.); i.a.safiullin@mail.ru (I.A.S.); adelnadeev@rambler.ru (A.F.N.); alex.a.korobkov@rambler.ru (A.A.K.)

* Correspondence: artur.k.gaysin@gmail.com

**Abstract:** Modern mobile networks exhibit a complex heterogeneous structure. To enhance the Quality of Service (QoS) in these networks, intelligent control mechanisms should be implemented. These functions are based on the processing of large amounts of data and feature extraction. One such feature is information about user mobility. However, directly determining user mobility remains challenging. To address this issue, this study proposes an approach based on multi-linear data processing. The user mobility is proposed to determine, using the multi-linear data, about the changing of the Signal-to-Interference-plus-Noise-Ratio (SINR). SINR varies individually for each user over time, relative to the network's base stations. It is natural to represent these data as a tensor. A tensor-based preprocessing step employing Canonical Polyadic Decomposition (CPD) is proposed to extract user mobility information and reduce the data volume. In the next step, using the DBSCAN algorithm, users are clustered according to their mobility patterns. Subsequently, users are clustered based on their mobility patterns using the DBSCAN algorithm. The proposed approach is evaluated utilizing data from Network Simulator 3 (NS-3), which simulates a portion of the mobile network. The results of processing these data using the proposed method demonstrate superior performance in determining user mobility.

**Keywords:** O-RAN; RIC; multi-dimensional data; tensor-based data processing

## 1. Introduction

Modern mobile networks require not only high data rates and low latency, but also the capability to serve the heterogeneous communications on demand. For example, Quality of Service (QoS) parameters significantly diverge for Enhanced Mobile Broadband (eMBB), Ultra Reliable Low Latency Communications (URLLC), and Massive Machine-Type Communications (mMTC). It is hard to carry out the conflicting requirements within a single physical network infrastructure.

The mobile network of a single provider can exhibit a complex, heterogeneous, and hierarchical structure. Moreover, this network should support a set of different Radio Access Technologies (RATs). Furthermore, User Equipment (UE) should support different standards, including 5G New Radio (5G NR), Long Term Evolution (LTE), and Wi-Fi. After all, the density of base stations' installation is also increased.

5G NR provides enhanced capabilities for wireless communications geared towards heterogeneous traffic. To achieve this, Network Slicing is used. This technology enables the creation of independent logical networks on a common physical infrastructure with optimized parameters for specific services and applications. Intellectual closed-loop management is implemented to control these kinds of mobile networks. However, this approach presents implementation difficulties due to the closed nature of the equipment interfaces provided by vendors.

The Open Radio Access Network (O-RAN) Alliance proposes to change this paradigm and introduces open interfaces between the elements of the cellular networks [1]. Moreover,

networks deployed under the O-RAN architecture consists of multi-vendor equipment and third-party software solutions. To control and manage the network parameters, programmable RAN intelligent controllers (RICs) are used [2].

The automatic optimization of QoS and the Quality of Experience (QoE) are the main functions of the RIC. The centralized storage of network data in the RIC allows us to adopt novel network management approaches. For instance, the Traffic Steering (TS) function facilitates user switching between cells for throughput maximization. RIC collects a large amount of data about network configuration, operating frequencies, cell and traffic parameters, users' mobility, etc. [3].

RIC algorithms are based on Artificial Intelligence (AI) and Machine Learning (ML) workflows [4]. Different approaches for AI and ML in RAN can be used, as well as those considered in [5].

Moreover, the problems of compromise between the secrecy rate and the consumed power are also important for RAN that can be considered as heterogeneous networks. To address this issue, the authors in [6] proposed the secrecy energy-efficient hybrid beamforming (BM) schemes for one type of heterogeneous network: the Satellite-Terrestrial Integrated Network (STIN). The concept of STIN was extended to the Hybrid Satellite-Terrestrial Relay Network (HSTRN) in [7]. Reconfigurable Intelligent Surfaces (RISs) were proposed for providing the required QoS for blocked users. To achieve this, the base stations of HSTRN in cooperation with RIS are utilized to enforce the satellite signals at the blocked users. In [8], the Malicious RISs are considered as green jammers in the Internet of Things (IoT) networks.

Moreover, the handover procedure is crucial in RAN, and significantly impacts QoS. Furthermore, UEs can be switched between different RATs in heterogeneous networks. Therefore, the standard handover algorithms between cells that select the serving cell may have a negative effect on QoS in networks with a complex hierarchical structure. The mobility pattern of the users can be utilized as a criterion for serving policy in heterogeneous networks.

In Self-Organizing Networks (SONs), LTE and 5G NR, the standard handover algorithms, are modified to solve the problems of the mobility procedures' stability and load balancing [9]. To this end, the handover parameters are adaptively adjusted. For example, the threshold level of the receiving signal or the hysteresis characteristic can be modified to enhance QoS during handover.

In [10], the authors proposed to improve the handover algorithm by employing the geographic coordinates of users. However, this method needs to collect and process data regarding user coordinates. Therefore, the indirect methods of evaluating user mobility can be utilized for more effective selection of the serving cell, serving policies, and making decisions about the handover.

The objective parameter that characterizes the quality of the radio channel is the Signal-to-Interference-plus-Noise Ratio (SINR). Additionally, modulation and coding schemes are chosen based on the SINR level [11] and can indirectly represent spectrum efficiency. Moreover, throughput in modern networks is primarily dependent on SINR rather than on the number of available channels. Furthermore, the change in the SINR over time, with respect to the serving base station and potential serving base stations, indirectly indicates the user movement pattern. Therefore, SINR can be employed to determine user mobility and enhance the handover algorithm.

However, SINR significantly varies due to changes in the real channel parameters. Moreover, SINR must be evaluated relative to the serving base stations and potential serving base stations. Consequently, these data are multi-linear and have a large volume. This amount of data cannot be processed using the classical methods of classification or clustering. Therefore, to overcome the curse of dimensionality, tensor-based processing can be employed. To this end, the multi-linear data are represented as multi-way tables and decomposed into matrices with fewer dimensions. This approach also allows us to decrease the noise and extract the features from observed multi-linear data.

In this paper, we analyze the users mobility pattern determining as a function of the RIC. This information can be used for more adequate Policy Management, ultimately enhancing overall QoS in the RAN. Drawing from the findings of studies [12–14], which focused on analyzing human mobility patterns, we can deduce that most individuals move according to distinct patterns and trajectories shaped by the urban environment's topography, such as major streets, park areas, and road junctions. For instance, most personal and public transportation subscribers navigate along predetermined routes, in alignment with the road network's structure and traffic regulations. Therefore, in our work, the necessary actions to ensure the desired communication quality (QoS, QoE) will be consistent for users with identical mobility profiles.

The key contributions of this paper can be summarized as follows:

1.   We propose the multi-linear model of the data that represent the SINR fluctuations and can be received from the RAN.
2.   We design a method based on the data decomposition and clustering to determine the user mobility profile within the RAN.
3.   Simulation results demonstrate that the proposed method effectively identifies user mobility.

The rest of the paper is organized as follows. Section 2 reviews the current state of the research field. Section 3 describes the bases of tensor algebra, the model of multi-linear data, and the proposed method for user mobility determination. In Section 4, we present the scenarios and simulation results. In Section 5, we discuss the limitations of the proposed method. Section 6 concludes the paper.

## 2. Background and Related Works

### 2.1. Architecture O-RAN

Traditionally, Radio Access Networks (RANs) are designed as monolithic structures, with each component functioning as a black box. Analyzing and tuning RAN parameters in such architectures is an impossible task.

To address this issue, as the opposite of complex and closed RANs, a new open architecture was proposed—Open RAN (O-RAN) [15,16]. The O-RAN architecture presented in Figure 1 is based on the following points: principles of the disaggregation of system elements, open interfaces between these elements, virtualization, and the programmability of elements. The O-RAN standardization ensures free interoperability between multi-vendor components. Moreover, the network functions of the O-RAN are software defined, virtualized, and implemented as white-box hardware.

In O-RAN, the NR Next Generation Node Base (gNB) is disaggregated into several nodes: Open Control Unit (O-CU), Open Distributed Unit (O-DU), and Open Radio Unit (O-RU). The O-CU contains the gNB radio resource management logic and performs IP protocol mapping of the UEs traffic. The O-DU implements the functions of the Medium Access Control (MAC) layer and performs scheduling functions to allocate the radio resources between traffic flows. The O-RU incorporates Physical (PHY) layer functions and performs radio frontend functions [4,17].

The O-RAN architecture includes two RICs for managing and controlling the RAN: Non-Real Time (non-RT) RIC and Near-Real Time (near-RT) RIC. The non-RT RIC optimizes the RAN parameters on a time scale exceeding 1 s, and includes the functions of the Service Management and Orchestration (SMO) framework. The near-RT RIC operates on a time scale between 10 ms and 1 s and controls the parameters of the O-DU and O-CU via E2 interface [4,17].

The RICs interfaces for interacting with RAN elements are defined in the O-RAN specifications. The controllers themselves are software-defined and consist of Application Programming Interfaces (APIs) and autonomous applications, known as xApps and rApps. The APIs provide message collection and routing functions as well as interface termination. The control logic is implemented using xApps and rApps in the near-RT RIC and non-RT RIC, respectively. Application autonomy refers to the fact that they can be third-party

developed and integrated into the RIC. Generally, xApps and rApps are AI/ML-based solutions capable of predicting network traffic and/or configuring RAN nodes based on current conditions [4,17].

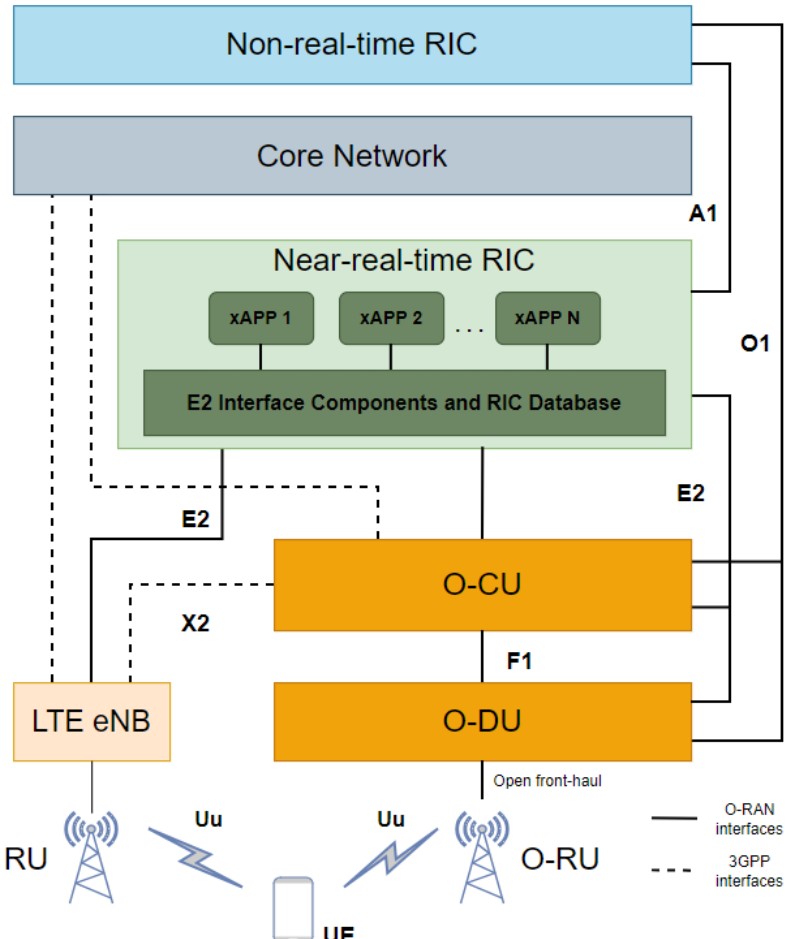

**Figure 1.** O-RAN architecture, with components and interfaces from O-RAN and 3GPP. O-RAN interfaces are drawn as solid lines, 3GPP ones as dashed lines.

Mobility-related metrics are employed in various RIC use cases, including Traffic Steering, QoE optimization, Handover, and Interference Optimization. These functions allow to automatically optimize the QoS and QoE. Therefore, the problem of the user mobility determining is considered in this paper.

### 2.2. User Mobility Pattern

One of the challenges in 5G NR systems is that signals at high carrier frequency exhibit poor penetration properties and high energy losses during reflection. Therefore, for this type of system, the transmitted energy should be concentrated in the user's direction, making user localization crucial.

On the one hand, ref. [18] presents a 3D positioning technique from the UE perspective. The core idea is to utilize massive MIMO systems in the BS, which allows the UE to realize high-precision positioning using a single antenna. Based on the sparsity of the mmWave channel, the CANDECOMP/PARAFAC (CP) decomposition is employed to estimate the positioning parameters (including the angles of departure and the time of arrival).

On the other hand, ref. [19] introduces a tensor-based 3D position estimation method for 5G massive MIMO systems. In this approach, the BS at a known position localizes the UE using the received signals from this UE. The authors employ the truncated Higher Order Singular Value Decomposition (HOSVD) model with dividing the receive antenna

array into the subarrays to estimate the angle of arrival (AoA), angle of departure (AoD) and delay of each part. Based on this information, the UE position can be estimated.

In contrast to [18,19], our proposed algorithm analyzes the user mobility rather than the user localization. The user mobility issues is explored in [20,21], where the modeling of user mobility in wireless communication is provided. The authors present a model that provides a realistic mobility model and verified the accuracy of their theory through extensive modeling.

In [22], the location and mobility management strategies are, considered. A new location management scheme, termed the mobility-pattern-based scheme (MPBS), is proposed to reduce the signaling traffic load and paging delay simultaneously. In [23], the user mobility in cellular networks is studied. The simple case of a single user traveling in a straight line within a single BS is considered, from which the probability distribution of path length without handover is derived. This knowledge can be used to improve the handover procedure.

Unlike the cited papers, the proposed method does not focus on user mobility modeling [20,21] or system efficiency improvement [22,23]. The primary motivation of this work is to close a knowledge-gap in the overall user mobility analysis. This will enable the separation of users into distinct groups based on their mobility patterns. This can be achieved because users move along certain trajectories determined by urban city structure [12–14], and users with similar mobility patterns can be represented as a single group. This group movement information can be used to ensure optimal policies and provide corresponding QoS for each user within the group. To cluster the users into groups, this paper considers SINR values as a signal that contains the user mobility information.

### 2.3. SINR Analysis

The analysis of the SINR values has been applied to address various issues, i.e., to improve bandwidth efficiency in [24], to reduce interference between femtocells and macrocells in [25], to increase capacity and improve energy efficiency in [26], and to produce the vertical handover in heterogeneous wireless networks in [27].

In [24], the authors propose an ML-based technique, i.e., an Artificial Neutral Network (ANN) to predict SINR in order to reduce the use of radio resources in 5G networks. Radio resource allocation is usually based on channel state estimation, i.e., SINR with the help of Sounding Reference Signals (SRS). To minimize the radio resource usage and hence to improve bandwidth efficiency, the Non-Linear Auto Regressive External/Exogenous (NARX)-based ANN is proposed, whose main objective was to minimize the rate of sending SRS.

In [25], results in game theory and stochastic approximation are used to investigate the problem of femto-to-mactocell cross-tier interference. As the main result, the authors propose an algorithm which maximizes the instantaneous performance of the whole system. The core of this algorithm is that all corresponding SINR observations of all active communications are available in both macro- and femtocells. Based on these observations, the femtocells learn the probability distributions of possible transmission configurations and perform the actions such that a minimum time-average SINR can be guaranteed in the macrocells, at the equilibrium.

The SINR analysis is extensively utilized in the development of new handover algorithms. An adaptive algorithm for handover in 5G and LTE-Advanced networks is presented in [28]. The algorithm relies on the SINR measurements to assess the channel state information. The authors do not consider the changes of the channel parameters over time; however, the adjustment of time-to-trigger is presented in algorithm.

A comprehensive survey of the handover management is presented in [29]. Improved classical and novel methods with AI/ML-based solutions are reviewed. When comparing input data for different methods, SINR is frequently utilized rather than Reference Signal Received Power (RSRP) or Reference Signal Received Quality (RSRQ). Experimental methods propose to extract the additional parameters related to mobility pattern, including the

velocity and moving direction of the users. However, it requires the additional hardware implementation beyond the existing mobile technology.

In this paper, the SINR values computed at several BSs located in the immediate vicinity of a given UE are considered as user information. In this case, when collecting such data for several dozen users, the problem of analyzing the received signals arises. One option for clustering is time series analysis [30,31]; however, to the best of our knowledge, the representation of correlated data in the form of a tensor gives more insight into the information contained in these data [32]. Thus, the acquired data will be represented as a tensor, and tensor decomposition techniques will be employed as a preprocessing step before clustering the users.

### 2.4. Tensor Decomposition in Wireless Communication

The use of tensor decomposition in wireless communications has its roots in [33]. The authors employ the parallel factor (PARAFAC) model in the direct-sequence Code-Division Multiple Access (DS-CDMA) systems to resolve a critical challenge: blind separation. To overcome the multi-user separation-equalization-detection challenge, the blind PARAFAC receiver is introduced, demonstrating a performance comparable to the non-blind receiver with minimum mean-squared error (MMSE).

This concept is further developed in subsequent articles. For instance, in [34], a PARAFAC-based approach is proposed, which leads to the solution of the simultaneous diagonalization problem. The authors of [35] propose an additional initialization step, which is called as enhanced line search with complex step (ELSCS), to improve the convergence of PARAFAC decomposition. This step is performed before Alternating Least Squares (ALS) algorithm which is used to compute PARAFAC decomposition. In [36], a constrained tensor model is introduced based on two constraint matrices defining the spatial spreading of the data streams and the spatial reuse of the spreading codes in multiple-antenna CDMA system.

Another field of research in wireless communication is channel estimation. To cope with this problem, tensor decompositions are applied in [37–42]. In [37], a joint channel estimation for a three-hop Multiple-Input Multiple-Output (MIMO) system with amplify-and-forward relaying protocol is performed. This estimator is based on ALS estimation by coupling PARAFAC and Tucker3 tensor models for the received signals to estimate the channel matrices in an iterative manner.

The authors in [40] investigate the issue of joint downlink (DL) and uplink (UL) channel estimation for millimeter wave (mmWave) MIMO systems using a tensor modeling approach. Assuming a closed-loop and multifrequency-based channel training framework, a two stage algorithm is proposed. In the first step, the joint estimation of the compressed DL and UL channel matrices is obtained in an iterative way (ALS algorithm) or in a closed-form solution. In the second step, different users' channel parameters are estimated (AoD, AoA, etc.) by solving independent compressed sensing problem.

In [42], time-varying and frequency-selective (TVFS) mmWave MIMO channels with high-mobility transceivers are explored. The frequency-domain received signal is represented as a third-order canonical polyadic (CP) model. Due to the nature of the TVFS channels, the factor matrices of the CP model contain the channel parameters, including AoD, AoA, time delay, path gain, and Doppler shift.

### 3. Model and Method

### *3.1. The Basis of Tensor Algebra*

We use the following notation in this paper. Scalars are denoted as lower-case italic letters $a$. Vectors and matrices are denoted by lower-case bold-faced letters (**a**) and upper-case bold-faced letters (**A**), respectively. Tensors are denoted as upper-case bold-faced calligraphic letters $\mathcal{A}$. Moreover, **a**($i$) defines the element ($i$) of a vector **a**. The same applies to a matrix **A** ($i, j$) and a tensor $\mathcal{A}$ ($i, j, k$).

The tensor $\mathcal{I}_{D,R}$ is $D$-dimensional super-diagonal tensor of size $R \times R \times \ldots \times R$. The element of this tensor is equal to one if all $D$ indices of this element are equal and zero otherwise. The $d$-mode product between a $D$-way tensor of size $M_d$ along mode $d = 1, 2, \ldots, D$ represented as $\mathcal{A} \in \mathbb{R}^{M_1 \times M_2 \times \cdots \times M_D}$ and a matrix $U \in \mathbb{R}^{J \times M_d}$ is denoted as $\mathcal{A} \times_d U$. It is computed by multiplying all $d$-mode vectors of $\mathcal{A}$ with $U$, whereas the $d$-mode vectors of $\mathcal{A}$ are obtained by varying the $d$-th index from 1 to $M_D$ and keeping all other indices fixed. The $d$-mode unfolding of the tensor $\mathcal{A}$ is denoted as $[\mathcal{A}]_{(d)} \in \mathbb{R}^{M_d \times M_{d+1} \cdot \ldots \cdot M_D \cdot M_1 \cdot \ldots \cdot M_{d-1}}$.

The canonical polyadic (CP) model of a $D$-way noiseless tensor $\mathcal{X}_0 \in \mathbb{R}^{M_1 \times M_2 \times M_3 \times \cdots \times M_D}$ is represented as

$$\mathcal{X}_0 = \Lambda_{D,R} \times_1 F_1 \times_2 F_2 \times_3 F_3 \times \cdots \times_D F_D, \tag{1}$$

where $F_d \in \mathbb{R}^{M_d \times R}(d = 1, 2, 3, \ldots D)$ are the factor matrices that are obtained after the factorization or CP decomposition of the tensor $\mathcal{X}_0$, $R$ is the order of the CP model or the rank of the tensor $\mathcal{X}_0$, and $\Lambda_{D,R}$ is a super-diagonal core tensor with loading factors $\lambda_r(r = 1, 2, 3, \ldots, R)$ on its super-diagonal that normalize the columns of the factor matrices to length one to avoid the scaling ambiguity of the CP decomposition. The CP decomposition of three-way tensor is illustrated in Figure 2.

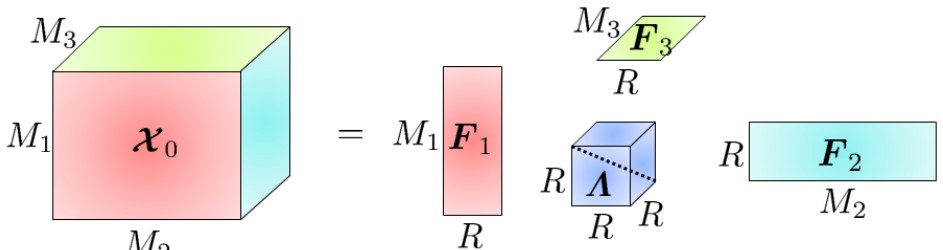

**Figure 2.** Canonical polyadic decomposition of a three-way noiseless tensor $\mathcal{X}_0$.

Thanks to the diagonality of the core tensor in the CP model (1), the factor matrices $F_d \in \mathbb{R}^{M_d \times R}(d = 1, 2, 3, \ldots D)$ consist of $R$ independent vectors that allow to separate and physically interpret the extracted components. This feature of the CP model is widely used in different areas like signal array processing and signal separation, as well as for MIMO systems [43] and biomedical applications [44]. Moreover, the CP decomposition allows to decrease the volume of the data that is a crucial point for our method of user mobility classifying, since the using of raw multi-linear data for considered problem is not suitable.

In practice, the recorded data are corrupted by noise. The tensor that is constructed from the noisy recorded data are defined as

$$\mathcal{X} = \mathcal{X}_0 + \mathcal{N}, \tag{2}$$

where $\mathcal{N} \in \mathbb{R}^{M_1 \times M_2 \times M_3 \times \cdots \times M_D}$ is the additive noise tensor. Therefore, (1) can be rewritten in the following way:

$$\mathcal{X} = \Lambda_{D,R} \times_1 F_1 \times_2 F_2 \times_3 F_3 \times \cdots \times_D F_D + \mathcal{N}. \tag{3}$$

Obviously, the rank of the noisy tensor $\mathcal{X}$ is not equal to the rank $R$ of the noiseless tensor $\mathcal{X}_0$. In general, it can be bigger. Therefore, the CP model of the tensor with observations $\mathcal{X}$ includes the estimates $\hat{F}_d \in \mathbb{R}^{M_d \times R}(d = 1, 2, 3, \ldots D)$ of factor matrices $F_d \in \mathbb{R}^{M_d \times R}(d = 1, 2, 3, \ldots D)$

$$\mathcal{X} \approx \Lambda_{D,R} \times_1 \hat{F}_1 \times_2 \hat{F}_2 \times_3 \hat{F}_3 \times \cdots \times_D \hat{F}_D = \hat{\mathcal{X}}, \tag{4}$$

where $\hat{\mathcal{X}}$ is a tensor that best approximates the noisy tensor $\mathcal{X}$.

The widely used algorithm for computing the CP decomposition is named Alternating Least Square (ALS) [45]. ALS is based on the solving of the linear least square problem

$$\min_{\hat{\boldsymbol{\mathcal{X}}}} \left\| \boldsymbol{\mathcal{X}} - \hat{\boldsymbol{\mathcal{X}}} \right\|_{\mathrm{F}}^2 \tag{5}$$

where $\|\cdot\|_{\mathrm{F}}^2$ denotes the Frobenius norm.

Fixing all but one factor matrices of the tensor $\hat{\boldsymbol{\mathcal{X}}}$ and sequentially solving the least-square problem (5) for each factor matrix, the best approximation of the tensor according to the CP model (4) can be computed. One of the next convergence criteria can be used as a stopping condition for solving the least-square problem: exceeding a specified number of iterations, the objective value is at or closed to the specified one, there are no significant changes in factor matrices or objective function.

As was mentioned before, the real data are corrupted by noise. The proper choice of the model order or the rank of the tensor with observed multi-linear data affects the accuracy, as well as the interpretability, of the results after the tensor decomposition. Moreover, the overestimating of the model order gives more noisy results with redundancy. Meanwhile, the CP decomposition with underestimated model order does not allow to extract full information from the multi-linear data.

In practice, the rank or the CP model order can be estimated by the visual inspection of the $d$-mode singular values profile of the tensor $\boldsymbol{\mathcal{X}}$. To this end, the Higher Order Singular Value Decomposition (HOSVD) is used also known as Multi-Linear Singular Value Decomposition (MLSVD) [46]. The HOSVD model of a tensor $\boldsymbol{\mathcal{X}}$ is defined as

$$\boldsymbol{\mathcal{X}} = \boldsymbol{\mathcal{S}} \times_1 \boldsymbol{U}_1 \times_2 \boldsymbol{U}_2 \times_3 \boldsymbol{U}_3 \times \cdots \times_D \boldsymbol{U}_D, \tag{6}$$

where $\boldsymbol{\mathcal{S}} \in \mathbb{R}^{M_1 \times M_2 \times M_3 \times \cdots \times M_D}$ is the core tensor and $\boldsymbol{U}_r \in \mathbb{R}^{M_d \times M_d}, (d = 1, 2, 3, \ldots D)$ are the unitary factor matrices. The $d$-mode singular values can be computed via the Singular Value Decomposition (SVD) of the $d$-mode unfolding of the tensor $\boldsymbol{\mathcal{X}}$ according to

$$[\boldsymbol{\mathcal{X}}]_{(d)} = \boldsymbol{U}_d \cdot \boldsymbol{\Sigma}_d \cdot \boldsymbol{V}_d^{\mathrm{H}}, \tag{7}$$

where $\boldsymbol{U}_d \in \mathbb{C}^{M_d \times M_d}$, $\boldsymbol{V}_d \in \mathbb{C}^{\tilde{M}_d \times \tilde{M}_d}$ are unitary matrices and $\boldsymbol{\Sigma}_d \in \mathbb{C}^{M_d \times \tilde{M}_d}$ is a diagonal matrix that has the $d$-mode singular values $\sigma_i^{(d)}$ on the main diagonal, $\tilde{M}_d = \frac{M}{M_d}$, and $M = \prod_{d=1}^{D} M_d$.

### 3.2. The Model of the Multi-Linear Data

In this paper, we consider the case when the direct navigate methods are not applicable. Therefore, the indirect methods and available data from the base stations should be used for determining the user mobility. To this end, we propose to analyze the changing of the SINR in time on the users side. The behavior of the SINR changing indirectly characterizes the user mobility. Note that the changing of the SINR is a very noisy parameter. Therefore, this fact should be taken into account when the method for processing this kind of data are designed.

Mobile users are served by different base stations. Therefore, we propose to consider the SINR changes of many users in time with respect to the base stations. Collecting these data during the relatively long time and stacking these data into the multi-linear table, the SINR changes can be natively represented as a three-way tensor. In summary, the three-way tensor $\boldsymbol{\mathcal{X}}$, which contains information about the SINR fluctuations, has dimensions cell ID $\times$ time $\times$ user ID and is expressed as a function $f_{\mathrm{SINR}}$ of three variables:

$$\boldsymbol{\mathcal{X}} = f_{\mathrm{SINR}}(\mathrm{CellId}, \mathrm{time}, \mathrm{UserId}) = \boldsymbol{\Lambda}_{3,R} \times_1 \boldsymbol{F}_1 \times_2 \boldsymbol{F}_2 \times_3 \boldsymbol{F}_3, \tag{8}$$

where $\boldsymbol{F}_1 \in \mathbb{R}^{CellId \times R}$, $\boldsymbol{F}_1 \in \mathbb{R}^{time \times R}$, and $\boldsymbol{F}_1 \in \mathbb{R}^{UserId \times R}$ are the factor matrices that include the information how the SINR is changed with respect to the base stations, time,

and users, respectively. The three-way tensor $\Lambda_{3,R}$ absorbs the weights of each vector from the factor matrices to avoid the scaling ambiguity of the CP decomposition. Figure 3 illustrates the CP decomposition of multi-linear observed data that is represented as a three-way tensor.

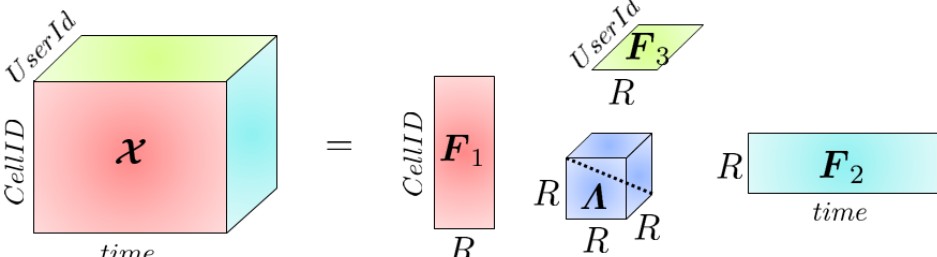

**Figure 3.** The CP decomposition of the 3-way tensor with the observed data.

The resulting factor matrices have significantly less size and can be analyzed for determining the user mobility. In real systems, the new obtained data can be represented as new slices or matrices and is added to the tensor $\mathcal{X}$. There is no reason to decompose this tensor again and the new factor matrices can be computed by representing the new data for each user as the linear combination of factor matrices and solving the following optimization problem:

$$
\begin{aligned}
&\underset{a}{\text{minimize}} && (S - \hat{S}(a)) \\
&\text{subject to} && \hat{S}(a) = \sum_{i=1}^{R} b_i \times \bar{a}_i \cdot \bar{c}_i
\end{aligned}
\tag{9}
$$

where $S$ is initial data, $\hat{S}(a)$ is approximate data which is represented as the linear combination of factor matrices $\bar{a}_i \cdot \bar{c}_i$ multiplied by the corresponded value $b$.

### 3.3. The Method

The proposed method includes the following steps. The received data from the base stations that include information about the changing of the SINR with respect to the base stations and time for each user are stacking into the three-way table or tensor.

In the next step, the model order of these multi-linear data are estimated. To this end, the singular values for each mode are computed according to (7). The number of dominant singular values determines the model order for the CP decomposition.

Next, the CP decomposition is performed according to (3). The resulting factor matrices include the information about the changing of SINR with respect to time, CellID, and UserID.

To separate users by their mobility, a variety of clustering algorithms, such as K-means, Gaussian Mixture Model, Agglomerative Hierarchical Clustering algorithm, Density-Based Spatial Clustering of Applications with Noise (DBSCAN), etc., can be employed. In this paper, the DBSCAN method is utilized for the automatic prediction of the number of clusters and the separation of the users by their mobility.

## 4. Simulations

The data for simulations were obtained from open-source discrete-time event simulator, namely Network Simulator 3 (NS-3). The scenario shown in Figure 4 describes the urban area, roads and crosses. In real-world scenarios, mobile base stations are placed near roads in areas with high traffic, such as highways or crossroads, to provide more robust service and improve network performance. In the presented scenario, we have also positioned base stations near roads to make the simulation more realistic. The distance between base stations is approximately 100 m, which is typical of a small cell scenario. The center frequency of the radio signal is set to 3.5 GHz, and the bandwidth is 20 MHz. The

3rd Generation Partnership Project (3GPP) Urban micro cell model is chosen as the channel model in NS-3.

User mobility is simulated as transport mobility, where users move along roads according to predefined trajectories. In this case, three groups consisting of five users each (red, orange, and green) are added to the scenario. Users in each group follow the same simulated path. At intersections, users can change direction, and in some parts of the route, users from different groups may move along the same section of the road simultaneously. To make the scenario more realistic, the starting and ending points, as well as the turning points at intersections, are different for each user. This ensures that each user has a unique trajectory while following similar patterns. Additionally, each user has its own channel model to ensure that each user experiences varying SINR values. The speed of each user is in the range of 30 to 40 km per hour, varying depending on group and section of the route. The second scenario has the same parameters as the first, but five randomly allocated static users are added (black).

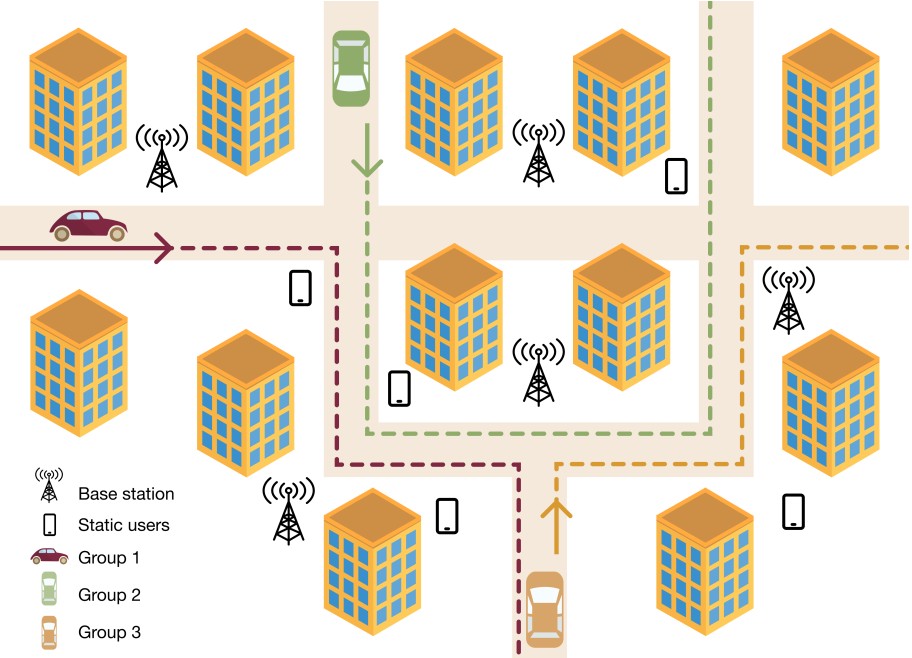

**Figure 4.** The scenario of simulation.

The obtained data from NS-3 contain the following parameters: timestep, time since the beginning of simulation; SINR, Signal to Noise and Interference Ratio; CellID, ID of measured cell; UserID, ID of measuring user. Periodicity of measurement is about 100 ms and SINR is written in double float format. Both scenarios simulate user movement for 60 s.

The obtained NS-3 multi-linear data are stacked in to the tensor with dimensions $15 \times 600 \times 5$ according to the model (8). In the first step of our proposed method, the model order is estimated. To this end, the $d$-mode singular values are computed (7). In Figure 5, the obtained $d$-mode singular values from the multi-linear data are depicted. According to the obtained results, the estimated rank of the tensor is 4.

In the next step, the CP decomposition is performed according to (4) and (5). The resulting factor matrices for the scenario with static users are depicted in Figure 6. The top plots in Figure 6 illustrate how the SINR between each user and all base stations is changed. The middle plot shows the evaluation of the SINR in time. The bottom plot represents the SINR changing for each base station. Moreover, in Figure 7, the factor matrices after the CP decomposition for the scenario without static users are depicted.

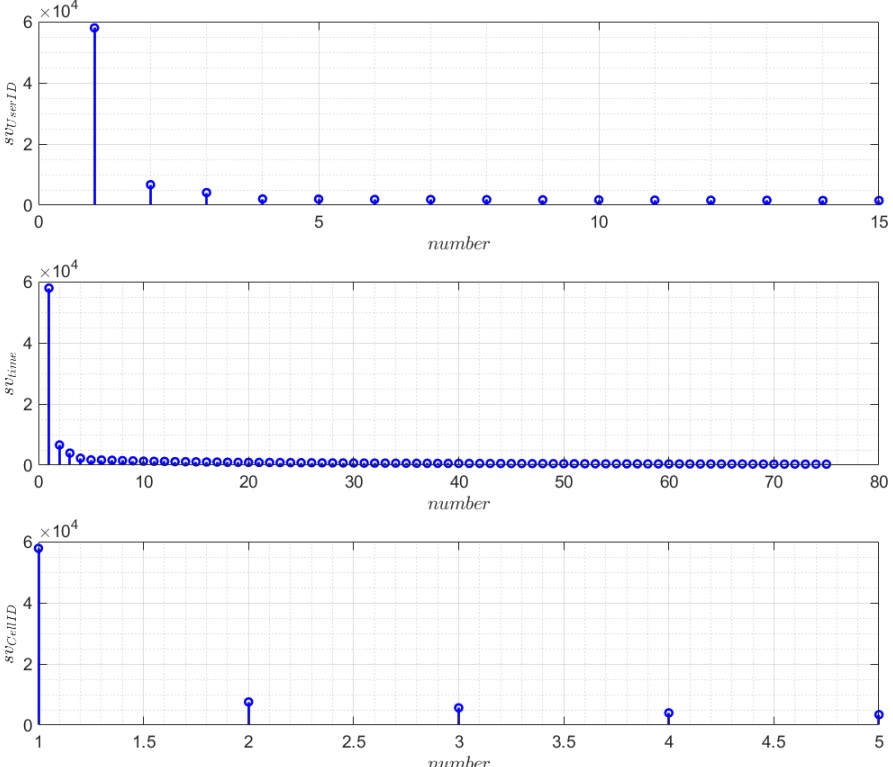

**Figure 5.** The *d*-mode singular values computed according to the (7) of the 3-way tensor with the observed data.

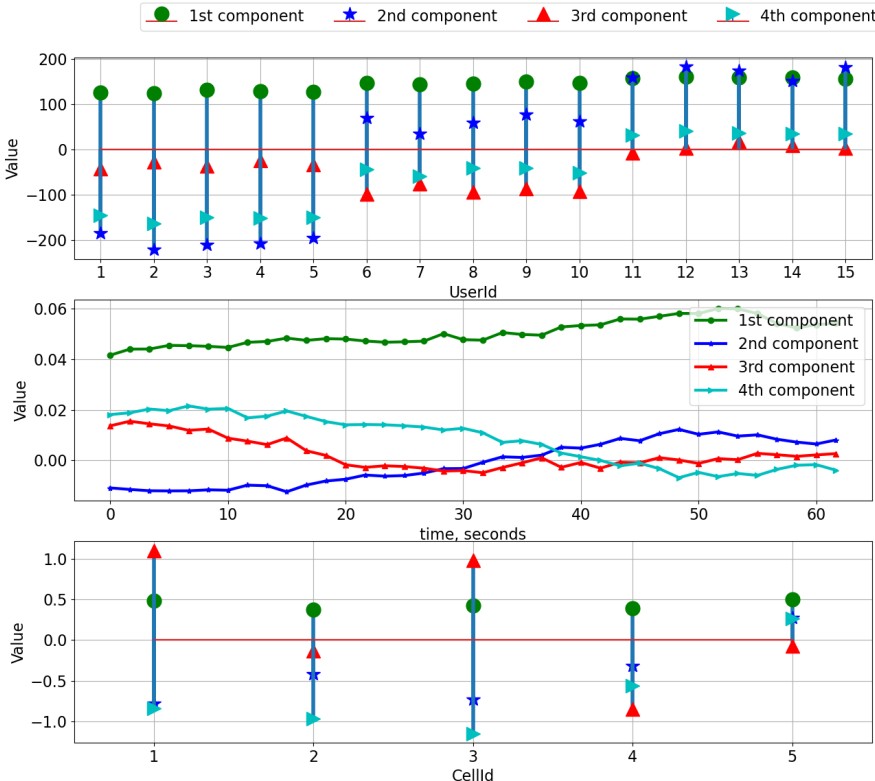

**Figure 6.** The factor matrices after the CP decomposition for the scenario without static users.

In the third step, clustering is performed based on the DBSCAN algorithm to determine the moving users and the patterns of their movement. To avoid introducing bias due to the different scales of the factor matrices, the factor matrices are normalized. Moreover, the outer product of the first and third factor matrices is used as input to the clustering algorithm.

Thanks to the DBSCAN algorithm, the number of clusters is automatically estimated. To this end, the parameters of the DBSCAN algorithm are tuned. The resulting parameters are: epsilon, 0.3; minimum number of samples, 3.

In Figure 8, the results of clustering are presented. As the initial group affiliation is known the accuracy of clustering could be determined. On the left side, results of DBSCAN for the first scenario is presented. Here, three clusters and all users from the same group are grouped to one cluster with no miss-clustered points. This means that five users belonged to one group according to the scenario, and are recognized as one cluster.

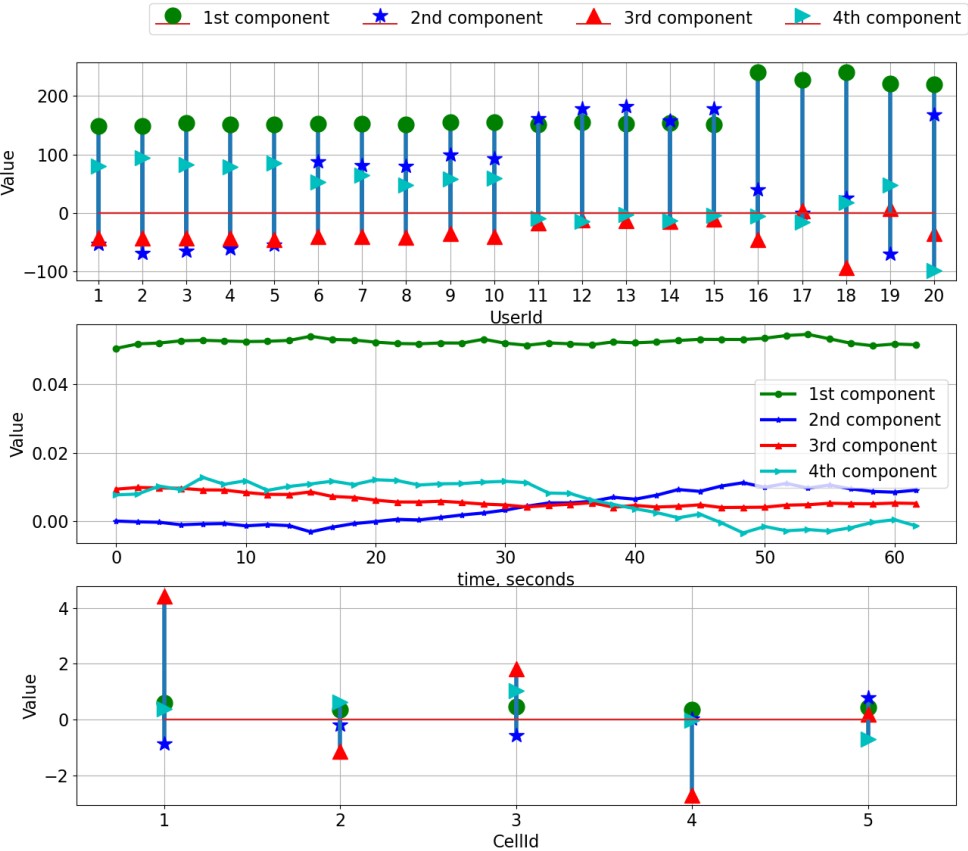

**Figure 7.** The factor matrices after the CP decomposition for the scenario with static users.

On the right side, we can see the results for the second scenario. Here, all users from groups are also recognized as belonging to one cluster, and all five static users do not belong to any cluster and are recognized with DBSCAN as noise points. In this case, the proposed method cannot separate them, due the user's random allocation and lack of similar patterns in SINR changing.

For both scenarios, the algorithm works well and the accuracy on the synthetic data for determined scenario is 100%. For a future step, we are planning to complicate scenario to research the limits of applicability of this solution.

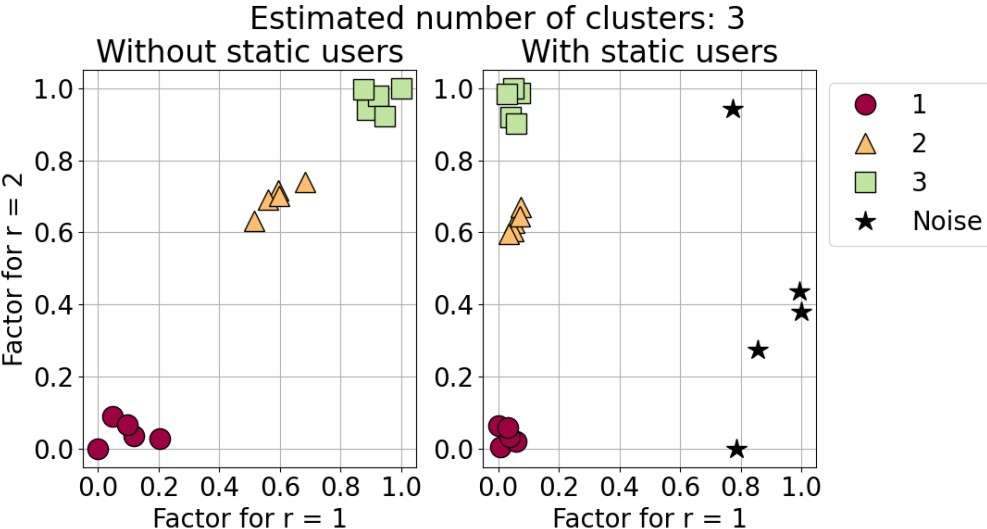

**Figure 8.** The results of clustering.

## 5. Discussion

Analyzing the results, the following limitations of the proposed method can be identified. The proposed method in this paper is considered as a function of the RIC that serves only one cluster or group of base stations. For considering the proposed method in terms of serving a few clusters, further investigation should be performed.

The method may also struggle to differentiate between user groups with low mobility, due to the lack of common patterns in SINR fluctuations. To address this, increasing the data collection duration can provide more substantial information.

The next limitation is the computational complexity of the tensor decomposition, which does not allow us to obtain the factor matrices in real time. Hence, the initial decomposition is performed in non-real time, while the new data can analyzed as a minimization problem provided in (9). Moreover, the replicated results can be detected by computing the Euclidean distance between the clustered data points and can be dropped out.

## 6. Conclusions

In this paper, we introduce an approach to identify mobility patterns based on the analyzing Signal-to-Noise and Interference Ratio (SINR) values. The raw data received from the Radio Access Network (RAN) cannot be directly clustered due to its massive size, multi-linearity, and high noise levels. However, these data can be effectively represented as a tensor. The model order is estimated by analyzing the singular values profile, while the CP decomposition is employed for feature extraction. The obtained factor matrices are then clustered using the DBSCAN algorithm, which eliminates the need for predetermining the number of clusters.

To demonstrate the effectiveness of the proposed method, simulations were conducted using the Network Simulator 3 (NS-3). Two scenarios were simulated, and the resulting data were analyzed using the proposed method. The simulation results indicate that our method can accurately identify distinct mobility patterns among users. This approach can serve as a foundation for an xApp that processes data within the RAN Intelligent Controller (RIC).

In the future, the proposed method can be integrated with machine learning-based agents to enhance the overall performance of the mobile network. Additionally, the proposed method can be implemented as part of the RIC functionality and validated on a testbed that emulates real LTE and 5G NR heterogeneous networks.

**Author Contributions:** Conceptualization, A.F.N. and A.K.G.; methodology, I.P.A. and A.A.K.; software, I.P.A.; validation, I.A.S. and A.F.N.; investigation, A.K.G., I.P.A., I.A.S. and A.A.K.; writing—original draft preparation, A.K.G., I.P.A., I.A.S. and A.A.K.; writing—review and editing, A.F.N.; supervision, A.F.N.; project administration, A.K.G.; funding acquisition, A.F.N. All authors have read and agreed to the published version of the manuscript.

**Funding:** This research was funded by Russian Science Foundation, grant number 23-69-10084, https://rscf.ru/project/23-69-10084/, accessed on 5 October 2023.

**Data Availability Statement:** The data presented in this study are available on request from the corresponding author.

**Conflicts of Interest:** The authors declare no conflicts of interest. The funders had no role in the design of the study; in the collection, analyses, or interpretation of data; in the writing of the manuscript; or in the decision to publish the results.

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
