# Peer review of "An Approach for Using a Tensor-Based Method for Mobility-User Pattern Determining"

_inventions, doi:10.3390/inventions9010001_

Round 1

Reviewer 1 Report

Comments and Suggestions for Authors

Please see the attached comments.

Author Response

Dear reviewers, we responded to all the questions in one word document. Thank you a lot

Reviewer 2 Report

Comments and Suggestions for Authors

This paper presents an innovative approach to determine user mobility patterns in 5G networks, addressing a critical challenge in modern mobile network technology. The utilization of Signal-to-Interference-plus-Noise Ratio (SINR) data in conjunction with tensor-based data processing not only represents a novel approach but also showcases a deep understanding of the complexities involved in managing and interpreting large-scale network data. The paper's methodology, which incorporates Canonical Polyadic Decomposition for feature extraction and the DBSCAN algorithm for clustering, is both technically robust and well-justified. The introduction effectively sets the stage by outlining the current challenges and the significance of the proposed method in the context of handover algorithms and SINR analysis in mobile networks.  The subsequent sections on the model and methodology, as well as the detailed simulations using the NS-3 system, are meticulously presented. However, the simulations offer a limited scope on the benefits of the proposed method. In conclusion, this paper is a significant contribution to the field of mobile network technology and has the potential to influence future research and practical applications in 5G networks.

Comments on the Quality of English Language

The paper could benefit from a language review process

Author Response

(The authors gave the same response as above.)

Reviewer 3 Report

Comments and Suggestions for Authors

The paper focuses on improving the management of modern mobile networks, specifically 5G networks, using a tensor-based approach. The paper’s objective has been reached systematically; however, there are areas where the authors could improve before being accepted, such as:

1-     The abstract seems to lack a clear and coherent structure. It starts by discussing the context of 5G networks but quickly shifts into technical details without a smooth transition. This might be confusing for readers who are not already familiar with the topic.

2-     The introduction section does not clearly articulate the specific research gap the paper aims to fill. A clearer statement of what makes the proposed tensor-based method novel compared to existing methods would strengthen this section.

3-     Also, there is a lack of a clear connection between the presented background information and previous research. The introduction would benefit from a more detailed discussion of how this work builds upon or diverges from existing studies.

4-     The review presented in Section 2 might not be comprehensive enough. It primarily focuses on O-RAN architecture and tensor decomposition, but it may lack a broader view of related research areas, especially concerning user mobility patterns in 5G networks.

5-     The connection between the reviewed literature and the paper’s specific research focus is not strongly articulated. This makes it challenging to understand how the existing work directly influences or relates to the proposed research.

6-     The section reads more like a summary of existing literature rather than a critical analysis. There is limited discussion on the strengths, weaknesses, or gaps in the existing research, which is crucial for positioning the paper's contribution.

7-     Section 3 focuses on the theoretical and mathematical aspects but lacks a clear connection to practical applications. How can this method be applied in real-world scenarios, particularly in telecommunications?

8-     The paper could benefit from a clearer explanation of why specific methodological choices were made. For instance, why was Canonical Polyadic Decomposition chosen over other potential methods? What makes this approach most suitable for the problem at hand?

9-     The simulation setups are described, but there is a lack of detailed justification for the choices made in these setups. It's important to explain why specific scenarios were chosen and how they are relevant to real-world applications.

10-  The simulations are based on scenarios created in a network simulator. However, there is no comparison with real-world data, which is essential to understanding how the proposed method performs in practical, real-life situations.

Comments on the Quality of English Language

The authors should improve the English usage in the paper. This includes correcting grammatical errors, improving sentence structure, and using more concise language.

Author Response

(The authors gave the same response as above.)

Reviewer 4 Report

Comments and Suggestions for Authors

The paper is a very interesting one, focused on a well known problem in mobile networks. The references are good ones, combining the new and very new ones with classical papers or other published documents in the area. Moreover, the authors are proving a very good theoretical background.

The transition from the theoretical approach to the simulations is very abrupt. The simulated use case is presented, but it is not well underlined why and how it is a representative one for the analyzed theoretical approach. Also it is not very clear highlighted how the simulations results are explaining and underlined the theoretical ones.

Comments on the Quality of English Language

The entire paper must be checked for language accuracy point of view by a native of this language. There are several unclear phrases, for example the one in the rows 1-2 "These networks are designed according the principals of convergence networks and should give the best service to the users". What means "principals of convergence networks"? And this is only an example from the very beginning of the paper.

Author Response

(The authors gave the same response as above.)

Round 2

Reviewer 1 Report

Comments and Suggestions for Authors

The authors have well addressed all my concerns, no further comments.

Reviewer 3 Report

Comments and Suggestions for Authors

Thanks for considering all the given comments. 

Reviewer 4 Report

Comments and Suggestions for Authors

I appreciate the improvements added to the paper and also the rephrasing of the unclear parts of the text. Now it is significantly better and closer to the standards of a publishable paper.